# Imagined speech event detection from electrocorticography and its transfer between speech modes and subjects
Aurélie de Borman [1] ✉, Benjamin Wittevrongel [1], Ine Dauwe[2], Evelien Carrette[2], Alfred Meurs [2], Dirk Van Roost [3], Paul Boon[2] & Marc M. Van Hulle [1,4,5]

Speech brain-computer interfaces aim to support communication-impaired patients by translating neural signals into speech. While impressive progress was achieved in decoding performed, perceived and attempted speech, imagined speech remains elusive, mainly due to the absence of behavioral output. Nevertheless, imagined speech is advantageous since it does not depend on any articulator movements that might become impaired or even lost throughout the stages of a neurodegenerative disease. In this study, we analyzed electrocortigraphy data recorded from 16 participants in response to 3 speech modes: performed, perceived (listening), and imagined speech. We used a linear model to detect speech events and examined the contributions of each frequency band, from delta to high gamma, given the speech mode and electrode location. For imagined speech detection, we observed a strong contribution of gamma bands in the motor cortex, whereas lower frequencies were more prominent in the temporal lobe, in particular of the left hemisphere. Based on the similarities in frequency patterns, we were able to transfer models between speech modes and participants with similar electrode locations.

The importance of speech becomes particularly evident when it is impaired or lost, as is the case in patients suffering from dysarthria or anarthria. However, the absence of speech does not imply the absence of cognitive or language abilities[1]. Various assistive technologies have been proposed to restore or provide an alternative means of communication (for a review, see Schultz et al.[2]), including brain computer interfaces (BCIs) used, among others, to decode brain activity directly into speech. Studies relying on electrocortigraphy (ECoG) have reported impressive advances in decoding performed speech[3,4], perceived speech[3,5], attempted vocalized speech[6,7], attempted articulated speech[7–9], and mimed speech (articulation without phonation)[10,11]. However, the accurate decoding of imagined speech remains elusive although keyed to serve the speech-impaired with an alternative communication channel. Even though recent progress has been made with patients unable to produce intelligible speech, it remains a challenge to help those who completely lack residual movements, such as locked-in syndrome patients[12]. For patients suffering from neurodegenerative diseases, imagined speech would not depend on the stage of the disease, which is a serious advantage compared to models relying on "attempted speech" and thus residual articulatory movements which might gradually disappear. Imagined speech decoding is challenging exactly due to the absence of a behavioral output that could be used as ground truth to align the neural recordings across trials when training the decoder[6,13–15].

Dynamic time warping has been used to align imagined speech trials to corresponding performed speech trials, without knowledge of the exact timing of speech events[13,16]. However, it was shown that using the subject's actual vocal onset instead of the cue onset, speech decoding significantly improved[17,18], suggesting that imagined speech decoding could benefit from a method that provides precise on- and offsets. Inspired by voice activity detection (VAD) techniques, recent studies have proposed to detect on- and offsets from ECoG for performed speech[3,19,20], perceived speech[3] and attempted vocalized or unvocalized speech[6,8]. The detection of imagined speech events was only recently studied using sEEG[21]. To the best of our knowledge, imagined speech event detection from ECoG has not yet been attempted. However, the broad coverage offered by EcoG combined with recent advances in detecting other speech modes holds promising potential. This study aims to bridge the gap by introducing an imagined speech detector based on ECoG signals.

There is general consensus on the anatomical similarities between performed, perceived and imagined speech. However, the degree of overlap is not well understood[14,16,22], as evidenced by activations in dissociated

[1]Laboratory for Neuro- and Psychophysiology, KU Leuven, Leuven, Belgium. [2]Department of Neurology, Ghent University Hospital, Ghent, Belgium. [3]Department of Neurosurgery, Ghent University Hospital, Ghent, Belgium. [4]Leuven Brain Institute (LBI), Leuven, Belgium. [5]Leuven Institute for Artificial Intelligence (Leuven.AI), Leuven, Belgium. ✉e-mail: aurelie.deborman@kuleuven.be

regions, as well as the frequency bands that are involved. Studies demonstrated that high-gamma activity is predominantly present during performed and imagined speech along the perisylvian cortex, including the superior temporal gyrus (STG)[23] and the ventral motor cortex[24]. While high-gamma activity yields the best performance for performed speech decoding, recent studies have demonstrated that low-frequency content is beneficial for speech decoding from ECoG[5,11,18], although differences in discriminatory information contained in each frequency band remain poorly understood[8]. In addition to differences in contributing brain regions and frequency bands, the way imagined speech itself is produced is widely debated. Imagined speech could be a form of attenuated performed speech[21] or it could be evoked without a specific articulatory plan[16,25], by relying on higher-level linguistic representations[18]. In this study, we compared imagined speech to both performed and perceived speech (i.e., listening). We assessed the contribution of each brain region and frequency band to gain knowledge about the neural correlates and to shed light on the nature of imagined speech.

The observed differences in contributing brain regions and spectral bands question the feasibility of a successful transfer of performed or perceived speech decoding to imagined speech decoding. One notable exception is Martin et al.[16] where the decoder was trained on ECoG recordings of performed speech and tested on those of imagined speech. The inferior imagined speech performance was attributed to dissociations in the underlying brain regions. More recently, speech detection models were transferred between performed, mouthed and imagined speech with sEEG[21]. While a better transfer was observed from imagined to performed speech than in the opposite direction, the detection accuracy drastically decreased compared to the unimodal case. In addition, transferring models between participants is also of interest. Even though various methods for transferring data or models across subjects have been proposed for EEG-based BCIs (for a review see Jayaram et al.[26]), the variety in placement and number of implanted electrodes and the local nature of high-gamma ECoG activity [27] are tacitly assumed to preclude a similar approach. Makin et al.[4] challenged this assumption by using ECoG recordings during performed speech of one subject in order to further train the performed speech decoder of another subject. Whether transfer learning can be exploited to enable imagined speech detection by transferring models trained on performed or perceived speech, on the same or a different subject, remains an open question that we approached in this paper.

In this study, we thus analyzed ECoG responses to performed, perceived (i.e., listening) and imagined speech of short sentences in 16 refractory epilepsy patients with subdural implants in various locations of the left and right hemispheres. We focused on a simple but efficient model to detect speech events with a straightforward interpretation of the underlying features. We found that 10 of the participants exhibited significant imagined speech events and examined the frequency bands that contributed to the detection, from delta to high-gamma, and how these depended on speech mode and electrode location. While we confirmed the strong contribution of gamma bands to performed and perceived speech detection in the motor cortex, we also observed a significant gamma contribution for imagined speech detection. The latter, however, was absent in STG where instead the contribution from lower frequencies became more prominent. We also examined how well an event detector trained on one speech mode performs on another and from one subject to another (transfer learning), and found that subjects shared frequency patterns, albeit differently for the motor cortex and STG. These findings could not only deepen our knowledge of the speech network but also help in the implementation of brain-computer interfaces based on imagined speech.

## Results
### Overview of the pipeline
We recorded ECoG activity from 16 participants and used a stimulation paradigm in which a sentence was displayed after which the speech mode was instructed (Fig. 1a). We analyzed smoothed Hilbert envelopes of recordings filtered offline in 7 different frequency bands (Fig. 1b). For each

time point, a linear regression model predicted the presence or absence of a speech event based on the Hilbert envelopes of each frequency band in a 250 ms centered window. The contribution of each frequency band to the event detection was quantified in terms of activation patterns as described in Haufe et al.[28] (Fig. 1c, right panel). If available, we also compared the results of the models with the functional brain mappings acquired by the clinicians who used them to mark the brain regions for resection thereby avoiding the patient's eloquent cortex.

Given the absence of a behavioral output during imagined speech trials, we assumed that the onset and offset timings are similar to that of performed speech[21,25] but with some extra precautions. For each subject, the average performed speech onset and offset was determined and a larger onset delay and smaller speech time retained and used as surrogate speech labels (see Supplementary Fig. 1). Time points right after the go cue and before the end of the speech trial period were discarded to avoid mismatched labelings. As an additional test set (referred to as *passive* dataset in opposition to the aforementioned *active* dataset) for imagined speech detection, we also looked at the performance of the model in the period surrounding the apparition of the task cue (Fig. 1a).

### Speech detection performance
We first analyzed the performance of the single-electrode speech event detector. Hereto, for each trial, the accuracy was computed as the percentage of frames that were correctly classified. For each speech mode and electrode, we performed a leave-one-trial-out cross-validation and expressed the final performance as the accuracy averaged over trials. Figure 2 shows the 588 electrodes pooled from the 16 participants. The results for each subject can be seen in Supplementary Figs. P1–16 together with the best accuracy for each speech mode and the functional mapping. To compare to chance-level performance, a permutation test was performed for the best electrodes of each subject and speech mode. Labels for half of the trials (randomly sampled) were reversed and the leave-one-trial-out procedure was repeated 10,000 times.

For performed speech, 13 out of the 16 subjects had at least one electrode for which the detection performance was above chance level ($p < 0.01$, permutation test, see Supplementary Table 4); for perceived speech detection again 13 subjects, but not all the same, had at least one electrode for which performance was above chance level. Finally, 10 subjects had at least one electrode where imagined speech detection was above chance level (permutation test on both an active and passive dataset, both had to be significant). For 2 subjects (P4 and P7), the performance at the best imagined speech electrode in the window centered at the task cue was very poor. The position of these electrodes being very posterior, it is likely that the visual processing of the cue explains this poor performance.

The best performances for performed and perceived speech were achieved in areas close to the superior temporal gyrus (STG). The performance peaks at 93.23% for performed and 93.98% for perceived speech for one subject's electrode in the right STG (see P14). However, the same electrode does not perform well for imagined speech (50.04% accuracy), possibly because of the absence of auditory feedback. In contrast, another subject's electrode located in the left STG which performed well for performed and perceived speech (85.41% and 83.54%, respectively) was also the best electrode for imagined speech for that subject with an accuracy of 74.01% (see P9). Both participants P9 and P14 had a left hemispheric dominance for language (see Supplementary Table 1). Similar contradictory findings for STG were reported by Soroush et al.[21] when comparing best electrodes for imagined, mimed and performed speech.

Superior performance was also observed in the sensorimotor cortex. One electrode reached an accuracy of 87.55% for performed speech in a motor region involved in mouth and tongue movements (see P15). That same electrode was also the most efficient one to detect imagined speech with an accuracy of 75.03%. In another subject, the best electrode for performed speech was also the best one for imagined speech, with performances of 85.86% and 79.83%, respectively (see P1). The two mentioned

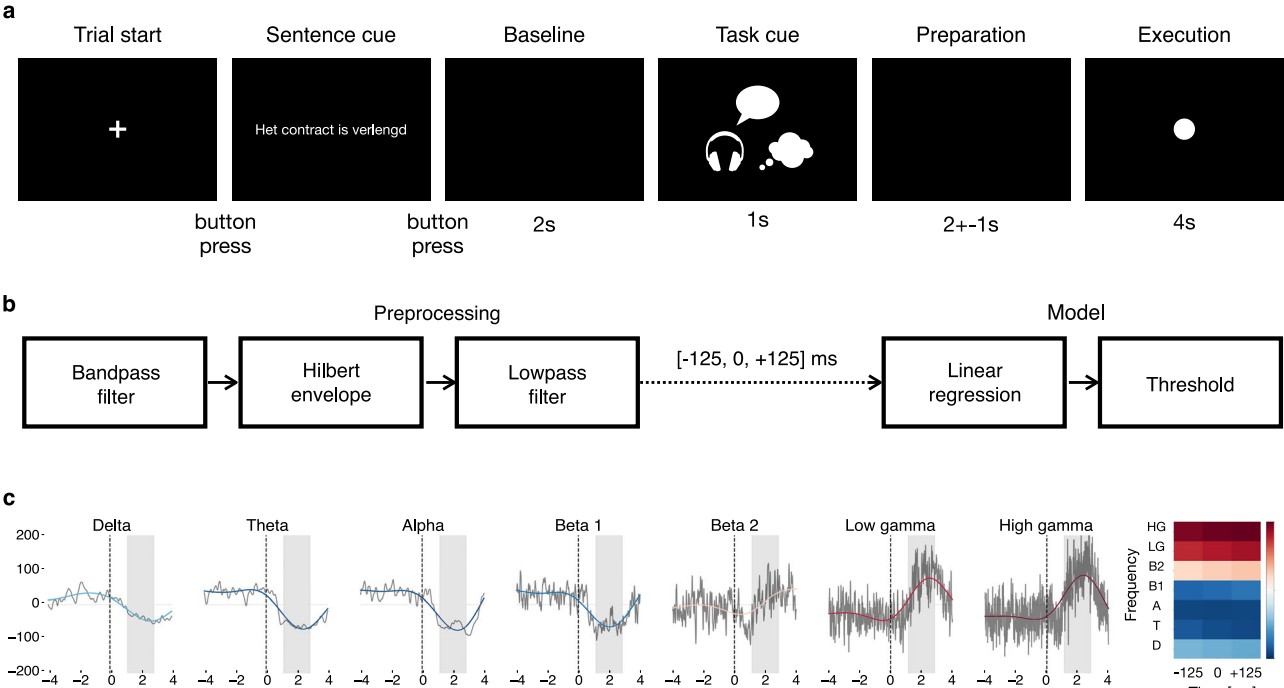

**Fig. 1 | Pipeline overview. a** Paradigm timing: The participant reads and memorizes the displayed sentence before pressing the space bar. One out of 3 task cues is then randomly presented to indicate the speech mode (performed, perceived or imagined) followed by a preparation window in which the large jitter is intended to suppress strong expectation activations. Finally, a circle appears (go cue) and the participant performs the task. **b** Speech detection pipeline: The ECoG signals are first band-pass filtered in 7 different frequency bands: delta (0.5–4 Hz), theta (4–8 Hz), alpha (8–12 Hz), beta1 (12–24 Hz), beta2 (24–40 Hz), low-gamma (40–70 Hz), and high-gamma (70–120 Hz). The Hilbert envelope is then computed for each band and the envelopes are low-pass filtered. Speech detection is performed at each time instance for which the features comprise the envelopes of each frequency band centered at that time instance (21 features in total). The constructed features are then fed into a linear regressor of which the output is thresholded. **c** Hilbert envelopes and activation pattern: The shown envelopes are averaged for one subject over trials (performed speech) for visualization purposes only. The dashed line indicates the go cue and the shaded area the window during which a majority of trials contained speech. The gray lines depict the Hilbert envelopes before low-pass filtering, the colored lines those after low-pass filtering. The colors of the lines correspond to the activation pattern depicted on the right for the 7 frequency bands (D = delta, T = theta, A = alpha, B1 = beta1, B2 = beta2, LG = low-gamma, HG = high-gamma) and 3 time instances.

**Fig. 2 | Speech detection performance. a–c** For each electrode, one third of the disk corresponds to a speech mode: performed (red), perceived (blue) and imagined (green). The radius is proportional to the accuracy averaged over all trials per speech mode; the scale is illustrated at the bottom for 50, 75 and 100% accuracy. Displayed electrodes are those of the 16 subjects, located on the surface of the left (panel **a**) and right hemisphere (panel **b**), and the medial view of the left hemisphere (panel **c**). For imagined speech, the performance is shown for the active dataset, but set to 50% if the performance on the passive dataset was lower than 50%. The results for panel a and b can also be viewed in the first column ("Full spectrum") of Fig. 4 for each speech mode separately. **d** The number of electrodes with an accuracy above 65% are shown for the three speech modes. For imagined speech, the performance on the active dataset was used, electrodes with an accuracy lower than 50% on the passive dataset were discarded.

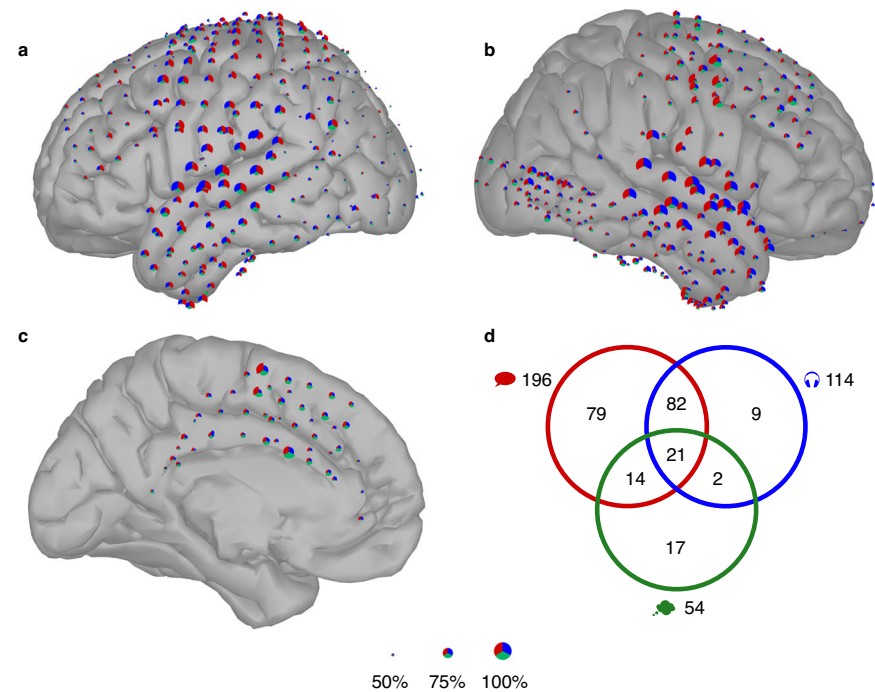

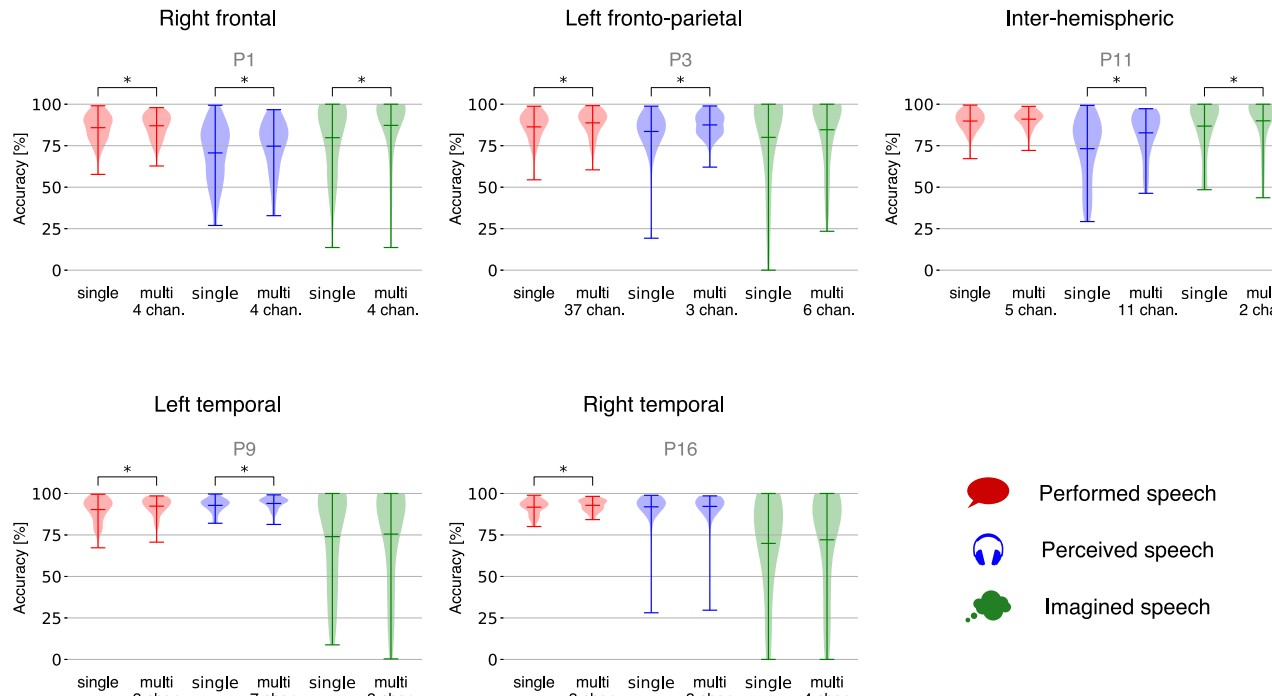

**Fig. 3 | Single- vs multi-electrode performance per speech mode.** Each panel corresponds to a single subject, labeled by the patient code. Five subjects were selected to illustrate the contributing brain regions (for plots of each subject, see Supplementary Figs. 3 and 4). For each subject, the performance for each speech mode is shown for both single- and multi-electrode models. Each violin plot depicts the distribution of trial accuracies. The middle line segment marks the mean while the upper and lower line segments mark the extrema. All single-electrode models perform significantly above chance level ($p$-value < 0.001, see Supplementary Table 4). The asterisk indicates whether the multi-electrode model outperforms the single-electrode model (*$p$-value < 0.05, one-sided Wilcoxon signed-rank test, see Supplementary Table 6). The optimal number of electrodes is listed below the multi-electrode performance.

electrodes were located in the dorsal part of the motor cortex of both subjects.

Interestingly, some electrodes located in the interhemispheric region also reached high accuracies. For one subject with interhemispheric electrodes only, the best electrode for performed speech was located in the right supplementary motor area (SMA), representing mouth and arm movements as established during functional mapping (see P11). That same electrode also performed well for imagined speech with an accuracy of 76.13% (second best electrode for that subject). Surprisingly, the overall top performance for imagined speech (86.81%) was achieved with P11 with an electrode in the cingulate gyrus.

Out of 588 electrodes (pooled across all subjects), 196 of them had an accuracy exceeding 65% for performed speech, 114 electrodes for perceived speech and 54 electrodes for imagined speech (see Fig. 2d). This highlights the overall sparser brain activation during imagined speech. Additionally, we noted that, although less electrodes were placed in the left temporal lobe, more electrodes reached a high accuracy for imagined speech detection in the left compared to the right temporal lobe (see Supplementary Table 5). A Wilcoxon rank-sum test confirmed that the accuracy in the left temporal lobe is larger than in the right temporal lobe ($p$-value = 2.34e−05, alternative: accuracy in the left temporal lobe is larger than in the right temporal lobe, $N = 140$ in the right temporal lobe and $N = 98$ in the left temporal lobe). It should be noted that the majority of the subjects were right-handed with a left hemispheric dominance for language (see Supplementary Table 1). We also observed a large overlap between the electrodes performing well for performed and perceived speech, with only a few electrodes specific to perceived speech. A larger number of electrodes were shared between performed and imagined speech in comparison to between perceived and imagined speech. However, we did not observe that the imagined speech electrodes were strict subsets of the performed or perceived speech electrodes, in contrast to Soroush et al.[21]. We also analyzed the Pearson correlation between the performance of imagined speech detection with

performed and perceived speech for all participants (see Supplementary Fig. 2). In both cases, we observe that imagined speech performance is positively correlated with performed and perceived speech performance (correlation of 0.4253 and 0.4086, respectively). However, when looking at the best electrodes only (performance above 65%), we observed that the trend is reversed for perceived speech (correlation becomes -0.3730), meaning that the best electrodes for perceived speech detection do not necessarily match with the best electrodes for imagined speech detection.

### Single- vs multi-electrode models

While it is possible to achieve a high accuracy with only one electrode, we tested whether the accuracy could be further improved by using multiple electrodes. The strategy was applied for an increasing number of electrodes, ranked by their Pearson correlation with the speech labels. We showcase here the performance for 5 subjects with different implanted regions (Fig. 3). In total, performed and perceived speech detection significantly improved for 10 subjects. In contrast, only 2 subjects had their imagined speech detection improved by the multi-electrode model. This is in line with the previous results showing that fewer cortical sites are activated by imagined speech[18,21,29].

### Full spectrum vs. low-frequency and gamma band speech detection

While typically only high-gamma activity was used to decode speech, lower frequency activity was recently shown to improve performance[5,18]. To assess the effect of lower and higher frequency bands on speech detection, the single-electrode speech detector was trained and tested with one of two subgroups of frequency bands: frequency bands from delta to beta2 (i.e., lower frequency speech detection) and combined low- and high-gamma bands (i.e., gamma band speech detection). The performance was then compared to that of the model using the full spectrum (one-sided Wilcoxon signed-rank test, $p < 0.05$). The results for all electrodes can be seen in Fig. 4.

**Fig. 4 | Speech event detection based on full spectrum compared to gamma and low bands.** Accuracies are shown for the model trained on the full spectrum, only gamma bands and only lower frequency bands (left, middle, right columns) for performed (panel **a**), perceived (panel **b**) and imagined speech (panel **c**). For each electrode located on the left and right hemisphere views, the radius is proportional to the single-electrode accuracy; the medial views are shown in Supplementary Fig. 5. Statistical tests (one-sided Wilcoxon signed-rank test, $p < 0.05$) were used to assess the performance of speech event detection when using only the lower bands or only the gamma band compared to the case when using the full spectrum. The white disks indicate whether the detector based on the full spectrum performs worse, black disks when the opposite is true, and the colored disks when there is no statistical difference. For imagined speech, the performance is shown for the active dataset, but the performance was set to 50% if for the passive dataset it was lower than 50%.

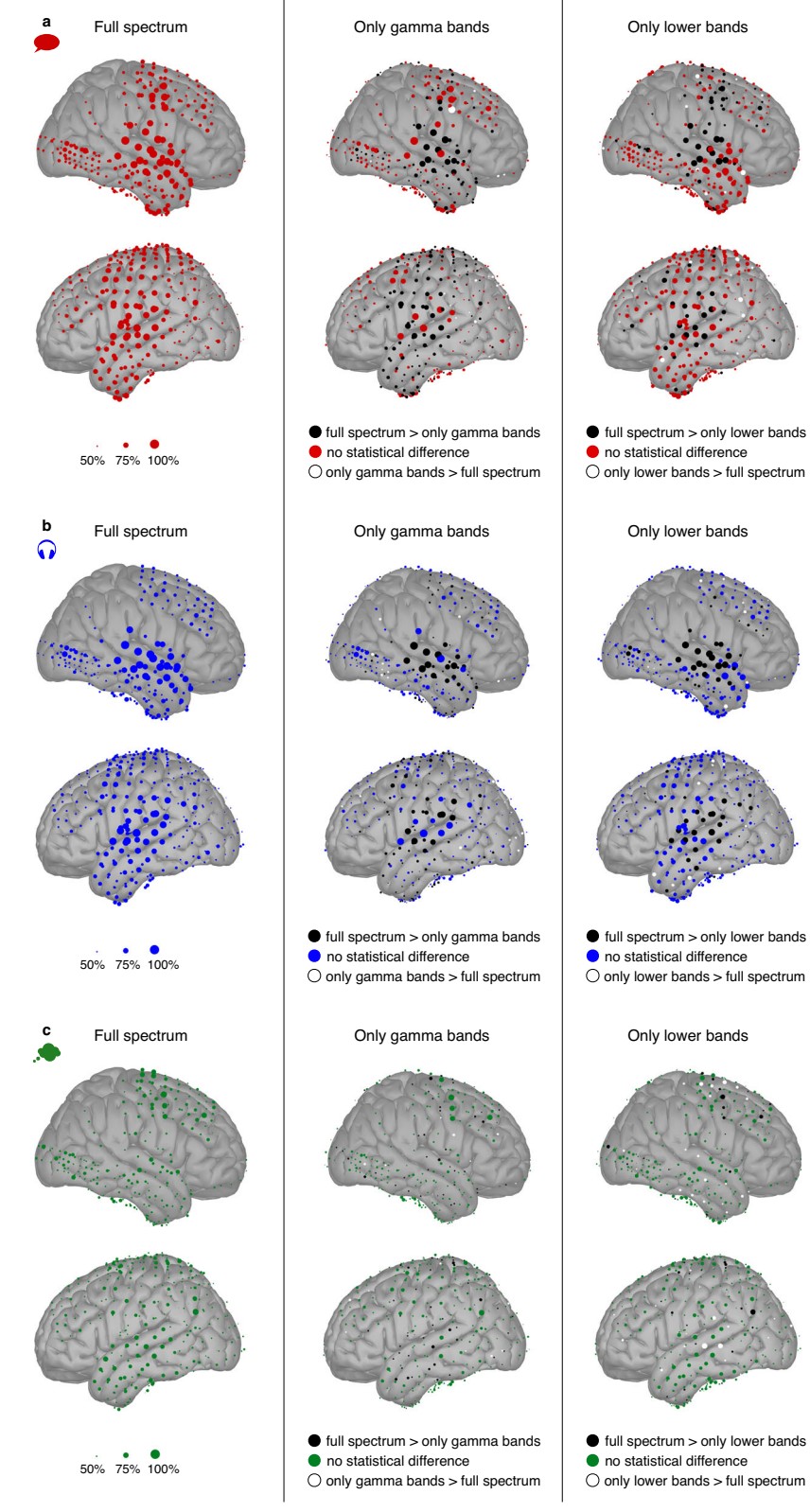

For perceived speech, the electrodes close to the primary auditory cortices could still perform well when using only the gamma band (although the performance was significantly lower than when using the full spectrum for most electrodes) while the other electrodes of the temporal lobes performed poorly. When only the lower frequencies were used, the performance of the electrodes close to the primary auditory cortices was significantly lower than when using the full spectrum but the performance in the rest of the temporal lobe remained stable, in contrast to the case when only gamma was used. This highlights the importance of lower frequencies in the temporal lobe, in particular in regions that are not close to the primary auditory cortices. A similar phenomenon was observed for performed speech in the temporal lobes. We conducted a Wilcoxon signed-rank test to compare the

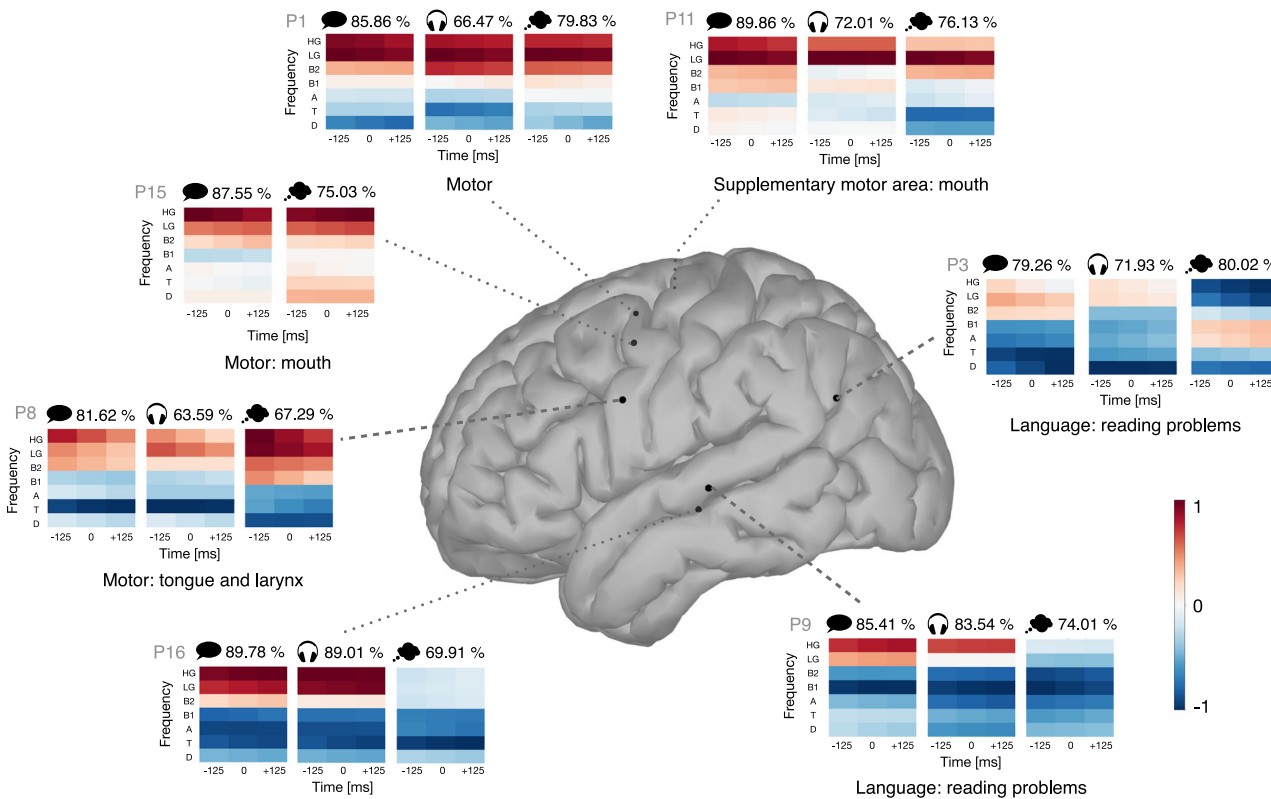

**Fig. 5 | Frequency band contributions per speech mode at the best imagined speech detection electrode.** The single-electrode accuracy and activation coefficients (color scale shown bottom right) are shown for 7 participants, for each speech mode (labeled by the icons), for the 7 frequency bands and 3 time instances. From our population of 16 participants, we display the results only for those participants for which the accuracy at the best imagined speech detection electrode was above chance level with $p < 0.001$ for all speech modes (see Supplementary Table 4). For subject P11, the second best imagined speech electrode (corresponding to the best electrode for performed speech) is shown since the functional mapping revealed that this electrode was located in the supplementary motor area while the first best electrode was labeled as 'seizure onset' (see Supplementary Fig. P11). The accuracies of the best electrodes for all subjects and speech mode can be found in Supplementary Figs. P1–16 and Supplementary Table 4. Dotted lines point to right hemisphere electrodes, dashed lines to left hemisphere electrodes.

performance using only low frequency versus only high frequency in the temporal lobe (alternative: accuracy using only low frequency > only high frequency) and the paired test was found to be significant ($p < 1e-7$, $N = 238$) for all speech modes, confirming the importance of lower frequencies in the temporal lobe. In the sensorimotor cortex, some electrodes could still perform equally well for performed speech using only gamma while others had a decreased accuracy compared to detection using the full spectrum. For imagined speech, the performance was significantly worse for most of the electrodes in the temporal lobe when using only gamma whereas the performance remained stable when using only lower frequencies. The results were more contrasting in the sensorimotor cortex. This provides evidence that the contribution of lower and higher frequencies depends on the brain region and speech mode[18].

### Comparison between speech modes

To shed light on how speech modes relate to each other, we took a closer look at the best electrodes for imagined speech detection of seven subjects (Fig. 5). Four of these electrodes were located in the motor cortex, two of them in the temporal lobe and one of them in the angular gyrus. The functional mapping showed either motor or language-related activity for each of these electrodes, except for one electrode for which no functional mapping information was available (although a nearby connection indicated sound perception, see Supplementary Fig. P16). For each of these electrodes and each speech mode, activation patterns and detection accuracies are shown (Fig. 5). Interestingly, we observe very similar frequency patterns for the 3 speech modes in the motor cortex. Even though perceived speech scored the worst in that region, the pattern was still similar

to that of performed and imagined speech, suggesting that the motor cortex is activated during perceived speech, although to a lower extent. In the motor cortex, the gamma bands contributed positively to the output of the speech detector (the activity increases during speech) while lower frequency bands contributed in an opposite way. The same pattern was observed for both performed and perceived speech in the temporal lobe. However, imagined speech did not elicit an increase of gamma activity in the temporal lobe, probably reflecting the absence of auditory feedback[18]. The patterns for performed and perceived speech were rather similar in the angular gyrus, although with a weaker activation of the gamma bands. In contrast, imagined speech exhibited a different frequency pattern in that region with a strong negative contribution of the gamma bands.

In order to further investigate the relation between speech modes and brain regions, we subdivided electrodes into three groups (temporal, frontoparietal and occipital regions) and conducted a few extra analyses based on this subdivision. Electrodes from the 10 subjects who had at least one electrode performing above chance level for imagined speech detection were included in this analysis (143 electrodes in temporal lobes and 232 electrodes in the frontoparietal region, see Supplementary Table 4). We defined as "gamma activation" the average of the activation pattern coefficients of low-gamma and high-gamma bands. We then compared the gamma activation for performed and imagined speech in both the temporal and frontoparietal regions. We observed a significant difference between the gamma activation of performed and imagined speech in the temporal lobe (Wilcoxon signed-rank test, $p = 1.42e-09$, alternative: gamma activation of imagined speech < performed speech, $N = 143$) while the $p$-value was much larger in the frontoparietal region ($p = 0.0377$, $N = 232$, see Supplementary Fig. 6).

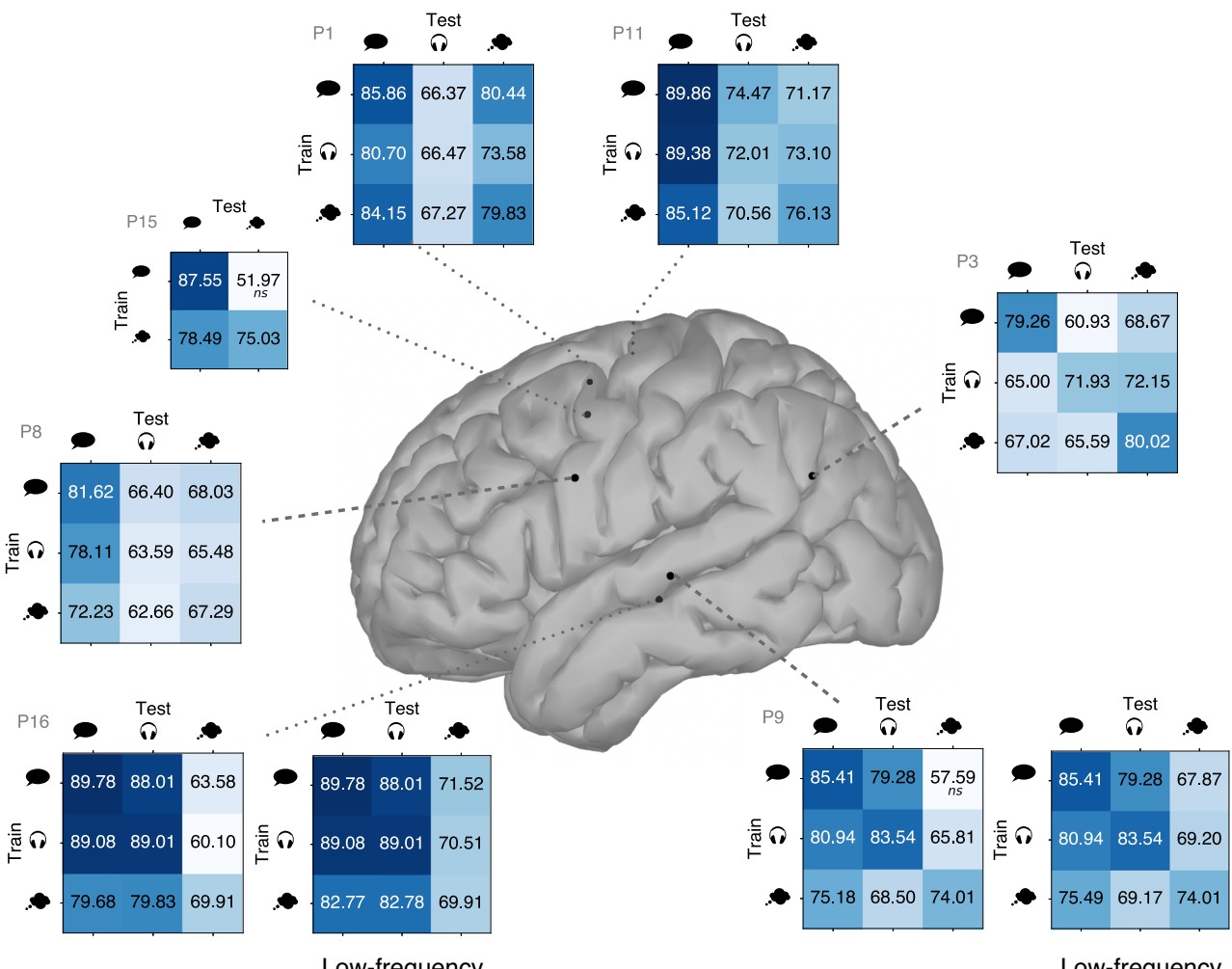

**Fig. 6 | Cross-modality transfer.** For each electrode, a train/test accuracy matrix is shown with single-electrode speech detection accuracies when a speech detector trained on one speech mode (rows) is transferred to the same or another speech mode (columns). The diagonal refers to the within speech mode accuracy. For the electrodes located in the temporal lobe, the matrix on the right of each pair lists the accuracies when only the lower frequency bands are used for a transfer from/to imagined speech; the transfer between performed speech and perceived speech remains unchanged. Accuracies that were not above chance level are marked with 'ns' (see the Methods section for the description of the test, $p < 0.01$ for all other cases, see Supplementary Fig. 9 for $p$-values).

To further investigate the difference between performed and imagined speech, we also conducted a statistical test to compare the drop in gamma activation from performed to imagined speech in both regions. The decrease in gamma activation was significantly larger in the temporal lobe than in the frontoparietal region (Wilcoxon rank-sum test, $p = 1.29e-04$, alternative: difference in the temporal lobe > difference in the frontoparietal region, see Supplementary Fig. 7). Finally, we also analyzed the relation between gamma activation and model accuracy in the temporal lobe (see Supplementary Fig. 8). The Pearson correlation was positive for both performed and perceived speech while it was negative for imagined speech, further indicating that gamma bands play a less important role in the temporal lobe for imagined speech detection.

**Transfer between speech modes**

Given that several electrodes of Fig. 5 share frequency patterns across speech modes, we hypothesize that speech detectors could be transferred between speech modes. Hereto, we trained a speech detector on one speech mode and tested it on another mode, in a way similar to Soroush et al.[21]. The obtained accuracies are reported in the matrices of Fig. 6. The performance remains quite similar in the motor cortex, except for the subject that had hearing problems (P15). In the temporal lobe, the transfer was not as successful as in the motor cortex: the transfer was only efficient between performed and

perceived speech. However, if instead of using the full spectrum, only the lower frequencies were used to transfer from/to imagined speech, given the lesser change in gamma activity (Fig. 5), a better performance was reached (additional matrices in Fig. 6). Transfer in the angular gyrus was not successful. These results add to the high similarity between speech modes in the motor cortex but less for the other regions.

**Transfer between subjects**

Thus far, our models were trained for each subject separately. However, we observed a certain similarity in the frequency patterns for subjects with similar electrode locations and we therefore attempted to transfer imagined speech detectors between subjects. We first trained the imagined speech detector on one subject and tested it on another subject (Fig. 7a). Subjects were divided into two groups, with an electrode on the motor cortex or on the temporal lobe, using the same electrodes as in Fig. 6 but omitting the one located in the angular gyrus, since it belongs to none of the two groups. Performance above chance level could be obtained for subjects with electrodes in similar brain regions. The highest accuracies resulted from a transfer within the same brain region compared to across brain regions, as expected from the similarities and differences in frequency band contributions (see Supplementary Fig. 11). Next, we also trained a model on subjects from the same group. The same proportion of each subject's trials

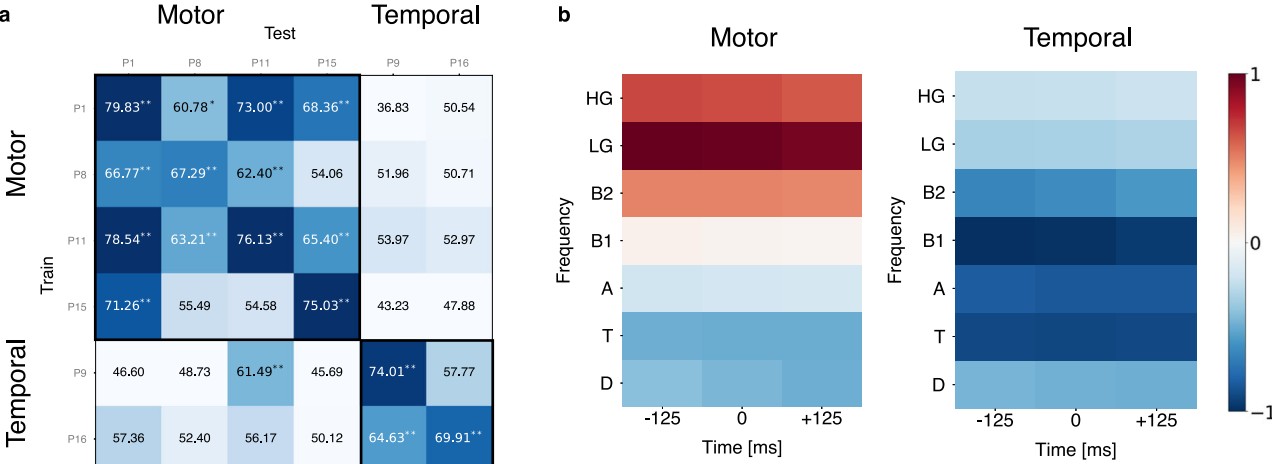

**Fig. 7 | Cross-subject transfer of imagined speech detection. a** The train/test accuracy matrix shows the single-electrode accuracies in transfers between subjects. Each row corresponds to the subject's data used for training and each column the subject on which the trained detector is tested on. The same electrodes as in Figs. 5 and 6 were used with the exception of subject P3 as the position of that electrode was neither frontal nor temporal. The asterisk indicates the outcome of the permutation test (\*p-value < 0.01, \*\*p-value < 0.001, see Methods section, see Supplementary Fig. 10 for p-values). **b** Frequency band contributions are shown for the population-based models. Activation coefficients are shown for the 7 frequency bands and 3 time instances used, for the motor cortex- (left) and temporal lobe models (right), using the same electrodes as in panel a.

were present in each fold of the cross-validation, both in the train and validation sets. The performance of imagined speech was neither improved nor decreased compared to the subject-specific performance ($p > 0.05$, Wilcoxon signed-rank test, see Supplementary Table 7). Figure 7b shows activation patterns for both models. The frequency patterns summarize the observations made in the previous sections: gamma bands are important in the motor cortex to detect imagined speech while lower frequencies dominate in the temporal lobe.

## Discussion

Attempts to align imagined speech events with ECoG[13,16,17] yielded limited success partly due to the jitter in imagined speech onsets across trials. In order to address the time-alignment issue, speech detection techniques have been applied to detect from ECoG performed-[3,19], perceived-[3] and attempted speech events[6,8]. To the best of our knowledge, there are no reports of imagined speech event detection from ECoG. Recently, Soroush et al.[21] proposed a model to detect imagined speech events from sEEG, yielding a 69% balanced accuracy (median of 7 subjects). Even though we did not have much recorded data per subject (60 trials per speech mode, which corresponds to a few minutes), we managed to obtain accuracies above 90% for performed and perceived speech. For imagined speech, detection was better than chance level for 10 out of 16 subjects, exceeding 75% in 4 subjects. In most cases, the best electrodes for speech detection matched the ones labeled as being involved in motor activity, audition or comprehension during functional mapping, providing evidence that the model relies on speech-related processes.

For two participants (P1 and P15), we obtained a high accuracy for imagined speech detection in the dorsal aspect of the motor cortex, similar to the optimal position for attempted speech detection in Moses et al.[6] Functional mapping confirmed for one participant that the electrode was located in a region involved in mouth motor activity although adjacent to the hand motor region[30], in line with recent studies decoding speech from the hand knob area[31,32]. For another participant (P11), an electrode on the supplementary motor area (SMA) also yielded a high accuracy for both performed and imagined speech detection, consistent with studies highlighting the role of this region in speech processing[33] in particular in sound imagination[34], one of the interpretations of imagined speech[17]. As for the temporal lobe, we observed that more electrodes performed well in the left hemisphere compared to the right hemisphere, with the best electrodes located over the STG. In general, we observed that fewer cortical sites were activated during imagined speech compared to the other speech modes as reported in

multiple studies[18,21]. Because of this, the multi-electrode model improved the imagined speech performance for only two participants while more participants benefitted from multiple electrodes for performed and perceived speech. This suggests that, only few electrodes might be sufficient to chronically implant for imagined speech detection and that high-density ECoG might not be required. This is in line with Vansteensel et al.[35] who implanted a patient with late-stage amyotrophic lateral sclerosis with low-density strips of subdural electrodes through burr holes. They could detect movement attempts using only one pair of electrodes (i.e., one signal given bipolar referencing). The location of the electrodes had been determined with the use of fMRI and anatomical landmarks. A similar procedure could be applied to detect speech attempts.

Linear regression has the advantage to have easily interpretable weights. Since quantifying the contribution of each feature based on linear model coefficients can lead to wrong conclusions, we relied on the procedure laid out in Haufe et al.[28]. While we were able to confirm a strong contribution of gamma bands to performed and perceived speech detection in the sensorimotor cortex, we also observed a significant gamma band contribution to imagined speech detection, with similar contributions from lower frequencies across the 3 speech modes. However, unlike performed and perceived speech, the low- and high-gamma contribution seemed absent for imagined speech in the STG. The lesser contribution of gamma bands during imagined speech in STG is in line with Proix et al.[18] who attributed it to the absence of auditory feedback in the imagined speech case. Martin et al.[17] also observed that some electrodes over the STG did not elicit a high-gamma increase during imagined speech although they did during performed and perceived speech. However, they were still able to decode imagined speech from high-gamma with other electrodes over the STG. Finally, we also observed an increased contribution from theta and beta activity during imagined speech, which has been attributed to the phonetic aspects of speech perception[36,37].

In this study, we observed that the frequency band contributions of imagined speech are more similar to those of performed speech in the motor cortex than to perceived speech in the STG. We also observed a larger overlap of electrodes between imagined and performed speech in comparison with the overlap between imagined and perceived speech. Hence, we believe that our results indicate that imagined speech is more likely related to imagined articulation rather than imagined audition. Furthermore, more electrodes were efficient in detecting imagined speech in the language dominant hemisphere (presumed to be the left hemisphere for right-handed

subjects[38]). A similar observation was made by Martin et al.[17] as imagined word pair classification performance was worse for right-handed subjects with a right hemisphere implant including temporal lobe coverage. The absence of audition, which activates bilaterally during perceived and performed speech, seems to cause an effect of hemisphere dominance on imagined speech detection. Importantly, it should be noted that the instruction might play a significant role in the execution of imagined speech. Participants in Proix et al.[18] were explicitly instructed either to imagine hearing (study 1) or to imagine speaking (studies 2 and 3). However, the authors found no strong evidence for any differences because of the limited overlap in the electrode coverage between the studies. In our study, we asked our subjects only to imagine speaking which might favor the activation of the motor cortex over auditory regions. In contrast, Martin et al.[17] instructed their participants to imagine hearing, which could explain the increased high-gamma activity in the STG (particularly prominent for the participant with the highest decoding accuracy) which is absent in our study. The effect of the adopted instruction warrants further investigation.

How transfer learning pertains to speech event detection is an open question. Interestingly, the frequency patterns of the perceived speech detectors were similar to those of performed and imagined speech detectors in the motor cortex. According to Cheung et al.[39], the motor cortex contains both sensory and motor representations, where the sensory representations are active during listening, whereas motor representations dominate during performed speech. Others have claimed that, when subjects listen to speech, they tend to mimic speech in their brain, activating motor areas[40]. Accordingly, the transfer between speech modes was shown to be particularly successful in the motor cortex while more precautions were required in the temporal lobes. The sparse coverage of the motor cortex in the study of Soroush et al.[21] probably explains why they observed a significant performance drop for transfers between imagined and performed speech. In the motor cortex, models trained on perceived speech could detect imagined speech with an accuracy above chance level for 3 subjects. This transfer is particularly interesting since training a model on perceived speech would be less tiring for the patient and perhaps lead to results faster than training on imagined speech. However, this also means that a model trained solely on one speech mode might produce unwanted output during other speech modes. This is likely to be the case for models trained on performed or perceived speech since auditory activity is predominantly present during both speech modes. The interplay between speech modes should therefore be kept in mind when developing a viable speech BCI. For imagined speech, the absence of gamma activity in the STG might be used as a discriminative feature to avoid spurious outputs due to perceived speech. Developing a single model that operates across the three speech modes is an interesting line of future research.

Finally, we also attempted to transfer models between subjects. We showed that it is possible to obtain an accuracy above chance level for imagined speech detection although the model was trained on another subject. This transfer was successful within specific brain regions although the electrode positions were not identical. This suggests that it might be possible to build a population-based speech detector. This is in line with Makin et al.[4] who observed an increased performance by pre-training a decoding model on another subject. It is likely that lower-level features (the presence or absence of speech) are transferrable across subjects while higher-level features (word discrimination) might not transfer so well due to inter-subject variability[18]. In any case, a population-based speech detector is particularly interesting for the implementation of a real-time interface. This would allow to provide immediate feedback, whether an attempt is detectable or not, which can be used in an operant condition setting to help the participant to improve his/her capability to imagine speech. Moreover, an efficient speech detector could alleviate the burden of paced cueing to converge faster towards a self-paced interface.

While the reach of this study was limited due to the restricted time for our experiments and the low-density electrode coverage, we believe our findings contribute to the implementation of imagined speech decoding pipelines. We acknowledge that, despite subjects were asked to refrain from any movement, non-perceivable muscle activity, even subliminal (e.g., vocal tract activity), cannot be ruled out without simultaneous electromyographic monitoring. Nevertheless, our results suggest that ECoG frequency bands in the temporal lobe and motor cortex contribute differently to speech events but consistently across subjects and speech modes, which could be exploited to pre-train components of the speech decoding pipeline.

## Methods
### Participants
For this study, 19 participants were recruited from Ghent university hospital (UZ Gent) after obtaining ethical clearance from the Commission Medical Ethics of UZ Gent. All ethical regulations relevant to human research participants were followed. All participants were epilepsy patients who were acutely implanted with subdural or epidural grids and/or depth electrodes as part of their clinical workup. Depth electrodes, if present, were not included in this study. Prior to their participation, all patients were informed about the procedures, and data processing and storage to which they were subsequently invited to give their written consent. Three participants reported they could not imagine speaking without moving their lips or producing a sound. Their recordings were not included in this study. For the one participant (P15) who was wearing a hearing aid, perceived speech trials were replaced by imagined speech trials, doubling the number of imagined speech trials. Clinical information about the participants can be found in Supplementary Table 1.

### Experimental design
Each trial began with the presentation of a fixation cross. The subject was asked to press a button when (s)he was ready to start the trial (Fig. 1a). Next, a short sentence was presented, which the subject was asked to memorize. In order to account for individual differences in memorization capabilities (due to age, education level, professional attainment, etc.), subjects were asked to press a button once they felt they memorized the sentence. After the button press, a black screen was presented for 2 s. Next, the task cue was shown for 1 s, informing the subject about the upcoming task (listening, performing or imagining speech). Following an empty screen, lasting 2 s with a maximal jitter of 1 s, a white circle appeared, and the subject was asked to execute the task corresponding to the cue (i.e., listening, speaking or imagining). All subjects completed three blocks, in each of which 20 short sentences were vocalized, imagined or perceived, corresponding to the task cue, presented in random order. Prior to the start of each trial, subjects were allowed to take a break. Due to this self-paced design, medical staff could attend to the patient at all times. The entire experiment lasted around 45 min. The 20 sentences (see Supplementary Table 2) were in Dutch (i.e., the patient's mother-tongue), were chosen from the LIST database[41], and were controlled for number of syllables and complexity. For the perceived speech task (i.e., listening), the auditory stimulation was controlled for intensity (60 dBA) and presented using a dedicated audio-interface (RME FireFace UC) and specialized in-ear headphones (Etymotic ER1). The experiment was implemented and presented to the participants using Matlab's Psychophysics toolbox for precise timing[42]. An experimenter was present throughout the whole experiment to check that the patient understood the instruction and would not produce articulator movements. If a doubt would have persisted, we could also ask to review the video sequence since patients are video monitored for clinical purposes. Finally, we also checked the audio recordings during imagined speech trials to check that no sound was produced. Three patients were excluded from the study since they could not imagine speaking without producing a sound or moving their lips (as mentioned in the previous paragraph).

### Data acquisition
The patient's ECoGs were recorded continuously throughout the experiment using SD LTM 64 Express amplifier (Micromed, Italy), operating at a sampling rate of 256 Hz. The platinum electrodes were embedded in silastic (Ad-Tech) and had a contact diameter of 4 mm (2.3 mm exposed) with 1 cm inter-electrode distance. The raw ECoG signals were checked for amplitude artefacts (signals should not exceed 1 mV) and contaminated

electrodes were removed prior to the analysis. No electrodes were removed due to interictal activity, given the absence of an objective criterion to do so. Nevertheless, the occurrence of interictal activity was assessed by an experienced epileptologist who provided level scores (level 1 = infrequent, level 2 = moderate, level 3 = frequent) to each channel which are reported in the supplementary material (see Supplementary Figs. P1–P16). Less than 10% of the channels exhibited frequent interictal activity (42 channels out of 588). Overall, electrodes with lots of interictal activity did not reach a high speech event detection accuracy. A notch filter was applied to remove the line noise. The subject's vocal response during the task window was recorded using an AT875R directional microphone (Audio-Technica, Japan), resulting in time-locked audio-recordings that were used for further offline analysis. Prior to the analysis, incomplete, interrupted or wrongly performed trials were excluded. The datasets were also checked for acoustic contamination[43] (see Supplementary Table 3).

## Localization of the intracranial electrodes
From the pre-implantation MRI scan of the participants, a cortical recon-struction and volumetric segmentation was performed using the FreeSurfer image analysis suite (version 6.0)[44]. The FreeSurfer output was then loaded into Brainstorm[45] and co-registered with a post-implant CT using the SPM12[46] extension. The coordinates of the implanted electrodes were manually obtained from the artefacts in the CT scan and projected on the cortical surface. All cortical visualizations were created using the Brainstorm toolbox[45] and custom Matlab scripts. Projection to a template brain (ICBM152) was also performed with Brainstorm.

## ECoG signal features
The EcoG recordings were re-referenced to a common average reference per intracranial grid. Next, the re-referenced signals were filtered in the delta (0.5–4 Hz), theta (4–8 Hz), alpha (8–12 Hz), beta1 (12–24 Hz), beta2 (24–40 Hz), low- (40–70 Hz) and high-gamma bands (70–120 Hz) using a 4th-order zero-phase Butterworth filter, and the Hilbert transform applied to extract the envelope for each of the frequency bands. The envelopes were then low-pass filtered below 0.25 Hz also using a 4th-order zero-phase Butterworth filter. This filter avoids the use of multiple smoothing/debouncing steps (e.g., by low-pass filtering of the Hilbert envelope, smoothing of the model output, and debouncing after thresholding, as done in Moses et al.[3]). However, the used filter could be adapted if shorter sen-tences were considered. For each time point, the prediction was based on the Hilbert envelopes at the time point of interest and two surrounding samples, at −125 ms and +125 ms (21 features in total, 7 frequency bands ×3 time points). Finally, the features were standardized (z-score).

## Speech features
From the audio recordings during the performed and perceived speech trials, the presence of speech activity was extracted by feeding the audio files and their transcriptions into a forced-alignment algorithm, which is part of a reading tutor algorithm[47]. This resulted in the extraction of the on- and offsets for each of the produced phones. All time-points between the onset of the first phone and the offset of the last phone were labeled as 'speech' while the others were labeled as 'no speech'. The on- and offsets were visually checked and manually corrected if necessary. This then serves as our ground truth against which model performance was judged. The dataset was balanced to contain as many 'speech' as 'no speech' time points. Hereto, for each trial, an equal number of samples were extracted during the speech window and during silence around the speech window (same amount before and after the speech window, except if not enough time points were remaining before the start of the next trial, then more samples were extracted before the speech segment).

## Imagined speech timing
For imagined speech, we hypothesized that the timing would be similar to that of performed speech (as done in Soroush et al.[21]). We adopted the same timing for all sentences as they were controlled for length. Trials were

aligned with respect to the go cue. In practice, we looked at the speech/no speech labels of the performed speech trials per time-point. Further, the speech/no speech labels were averaged over trials (in this case, 0/1 indicates the absence/presence of speech). The time window for which the averaged speech label was above 0.9 was selected as speech segment for the imagined speech trials (i.e, more than 90% of the performed speech trials contained speech during that window, see Supplementary Fig. 1). Note that, in this way, we overestimate the onset delay and underestimate the offset. As to 'no speech', time points were selected before the go cue and at the end of each trial if the averaged label was below 0.1. A maximum of 500 ms was selected at the end of the trial to favor the selection of data before the go cue for which the probability of containing imagined speech is almost zero. Time-points right after the go cue as well as right after the estimated speech window were not given any label and were discarded from the analysis as it is unclear whether they would exhibit imagined speech.

As to performed and perceived speech, the datasets were exactly balanced. Since we discarded some time windows, the dataset size was slightly inferior for imagined speech compared to the 2 other speech modes. Therefore, we also considered an additional test set. A dataset (referred to as *passive* dataset) was constructed by taking a 4 s window centered at the task cue. During that window, no intentional imagined speech was assumed to occur. This dataset was used to define an exclusion criterion to reject elec-trodes that are likely to respond to visual cues rather than to imagined speech events. If the accuracy on the passive dataset was below 50%, the overall performance was set to 50% or the electrode was excluded from the analysis depending on the context.

## Statistics and reproducibility
A leave-one-trial-out cross-validation was adopted to assess the perfor-mance of the event detectors. Hereto, the accuracy was computed for each trial as the percentage of frames correctly classified, and then averaged across all trials. Note that the accuracy was balanced due to the equal number of speech and silence time-points. In order to compare the averaged accuracy to chance-level performance, a permutation test was conducted for the best electrode and speech mode of each subject (because of the high computational cost). Labels for half of the trials (randomly sampled) were reversed (speech labels became silence labels, and vice-versa) and a leave-one-trial-out cross-validation was applied. This procedure was repeated 10,000 times. The p-value was equal to $(N + 1)/(10,000 + 1)$ where N is the number of times that the accuracy with the shuffled dataset exceeded the accuracy with the true labels. We did not shuffle the labels per time-point, but instead per trial, to keep a low-frequency input, which renders the test more conservative. Strictly speaking, the chance level is 50% since the datasets are perfectly balanced. To assess the performance of the transfer between speech modes and subjects, only labels of the validation sets were shuffled (per trial). When comparing two model accuracies (e.g. single- vs multi-electrode), we used a one-sided Wilcoxon signed rank-test with a significance level set to 0.05. The sample size for these tests is equal to the number of trials (see Supplementary Table 1).

## Single-electrode model
Linear regression was used to predict the time window during which speech occurs. Labels were set as −1 and 1 in the case of the absence and presence of speech, respectively. The Python package scikit-learn[48] was used to imple-ment the linear regressor. The output of the model was then thresholded at 0: a negative value was considered silence while a positive one speech. To study the contribution of the frequency bands, we computed the activation patterns as described in Haufe et al.[28]. Since we performed a leave-one-trial-out cross-validation, the same number of models as trials were obtained and activation patterns averaged across folds. The averaged coefficients were normalized so that the maximum absolute value of the coefficients was 1.

## Multi-electrode model
To combine information from multiple electrodes, models were first created for each electrode separately. Next, the continuous outputs were input to a

second regressor, to keep the dimensionality of the multi-electrode regressor low. Finally, the resulting continuous output was thresholded. The selection of the electrodes was based on the Pearson correlation between the features and the speech labels. For each electrode, we obtained 21 correlation values (one for each feature). The correlation coefficients were then averaged for each frequency band and the electrodes ordered according to the maximum correlation (in absolute value) amongst the 7 frequency bands. The ordering of the electrodes was defined with the correlations computed on the training set only. The accuracy as a function of number electrodes is shown for all subjects in the supplementary material (Supplementary Figs. 3 and 4) as well as the distributions of the trial accuracies for both the single-electrode and optimal multi-electrode models. We used statistical tests in a recursive manner to define the optimal multi-electrode model of which the performance is shown in Fig. 3, Supplementary Figs. 3 and 4. Initially, the best single-electrode model is considered as optimal. At each step, we add an electrode and check if the model performs better than the current optimal model (one-sided Wilcoxon signed rank test with alpha set to 0.05). If it does, the optimal model is updated. This recursive procedure favors a model with a small number of electrodes but should not be considered as rigorous as a held-out test set or nested cross-validation. If no multi-electrode model performed significantly better than the single-electrode model, the model with the highest accuracy was selected as the optimal one.

### Transfer between speech modes and subjects

When transferring a model between speech modes/subjects, the model was trained on the full dataset of the training speech mode/subject and tested on the full dataset of the tested speech mode/subject. Datasets were individually z-scored. We also built a model trained on a subset of subjects. In this case, we used a similar leave-one-trial-out procedure as explained in the previous section but using a validation set with one trial per subject. We took the same number of trials for each subject.

### Reporting summary

Further information on research design is available in the Nature Portfolio Reporting Summary linked to this article.

### Data availability

The datasets presented in this article are not readily available because of the sensitive nature of intracranial patient data. Requests to access the datasets should be directed to the corresponding author. Source data for Figs. 1 and 4 is available in Supplementary Data 1. Source data for Fig. 3 is available in Supplementary Data 2.

### Code availability

Code was written in MATLAB and Python and can be provided upon reasonable request. Please contact the corresponding author via email with any inquiries about the code.

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

## Acknowledgements
The authors wish to express their gratitude to J. Vanthournhout, E. Verschueren, E. Khachatryan, T. Francart and A. Kotarcic for their assistance and expertise. A.dB. is supported by the Research Foundation - Flanders (FWO 11K2324N). B.W. is supported by a post-doctoral mandate from KU Leuven (PDM/19/176). M.M.V.H. is supported by research grants received from Horizon Europe's Marie Sklodowska-Curie Action (grant agreement No. 101118964), Horizon 2020 research and innovation programme under grant agreement No. 857375, the special research fund of the KU Leuven (C24/18/098), the Research Foundation - Flanders (G0A4118N, G0A4321N, G0C1522N), and the Hercules Foundation (AKUL 043). A.M. is supported by research grants received from the Research Foundation - Flanders (G0A4321N, G0C1522N). The resources and services used in this work were provided by the VSC (Flemish Supercomputer Center), funded by the Research Foundation - Flanders (FWO) and the Flemish Government.

## Author contributions
B.W. and M.M.V.H. devised the experimental paradigm; I.D., E.C., A.M., D.V.R. and P.B. recruited the patients; B.W. and A.dB. recorded and analyzed the data; P.B., D.V.R., A.M. and M.M.V.H. supervised the study. All authors approved the manuscript.

## Competing interests
The authors declare no competing interests.
