## [Peer Review File · Communications Biology]

Reviewers' comments:

Reviewer #1 (Remarks to the Author):

commsbiol 23-3065

The manuscript reports on detection of speech spoken, heard or imagined, using ECoG data from 16 subjects, with a focus on imagined speech. Several types of analyses are presented, to assess relationships between the three. The logic of the chosen analyses and the nature of the results is not quite convincing, it seems more of a data exploration effort than testing clear hypotheses. Given the difficulty of obtaining the valuable human ECoG data, electrode location and patient are intermixed, resulting in challenges to separate regional phenomena from individual differences. Several aspects of the presented research dampen my enthusiasm, mainly because of a lack of second-level analyses to substantiate interpretations/conclusions, and the fact half of the analyses was conducted on an extremely small selection of electrodes. Accordingly, results are not very amenable to generalizable conclusions based on quantitative assessment, for which reason I hesitate to see how this moves the field forward. Yet I do think it is possible to improve the robustness of findings when second-level analyses are added and results thereof considered. In current form, there is no clear discourse on effect size of different data analysis comparisons, nor any second-level analysis probing consistency of effects across subjects. Furthermore, half of the analyses sections involve only 7 (or even 6) electrodes out of the total pool of 588, and conclusions are drawn wide based on this very small selection. It is also not explained why these electrodes (1 per subject for 7 subjects) were chosen. Any interpretation of differences between electrodes is thus quite possibly confounded with differences between individuals (all of whom have epileptic activity on top if it). Results are discussed and interpreted as more generally applicable than analyses support, given that there are no second-level assessments across subjects. Grids understandably differ in brain region coverage across subjects but at least one could form subgroups with similar coverage on particular brain regions for this purpose, allowing for more powerful second-level (across-individual) analyses

Comments

Methods/Results Section

Authors attribute much importance to the degree of low frequency contribution to speech detection and differences across brain regions. Yet, one possible explanation deserves consideration: results of separate bands for decoding speech regarding the gamma bands in auditory cortex may be explained by auditory attention processes increasing gamma power throughout the task, thereby increasing baseline activity and reducing any difference with further activation. This would be somewhat less in sensorimotor cortex since speed (hence preparation/readiness) is not important for participants. One could argue that the seen differences in high/low frequency contribution reflects functional differences and can therefore not be directly compared in terms of anatomical features between regions

In the comparison between speech modes section on p7, only 7 subjects and only 1 electrode per subject is analysed, out of 588 total. How is this motivated? Results are then hardly representative for a larger group, so the information obtained from this analysis seems marginal and conclusions rather overly generalized

Was each included electrode investigated for epileptic activity? given that grids are placed on clinically suspected epileptic zones, many electrodes could exhibit interictal phenomena contaminating the signals (hence analyses). For instance, for P3 a negative gamma response is observed for imagined speech, and virtually no response in the other 2 conditions. A negative gamma response may well be related to epileptogenic phenomena, where mental focus/attention tends to suppress the normally elevated interictal gamma power.

How was it determined that there was no residual articulator movement during imagined speech? such may well have generated sensorimotor cortex activation

What was the nature and purpose of the passive dataset analysis? As a sort of correction it seems, it was not applied to the other task types. Thresholds for effects in this dataset seem to be arbitrary as I miss a clear rationale

Was there an actual benefit from including separate time epochs (3 of 125 ms consecutive), given the graphs (fig 5 and 7) indicate virtually no differences? collapsing these 3 timepoints could add some power to analyses

fig 4: color coding is rather unclear (same basecolor for opposite effects), consider using different colors for larger/equal/smaller

fig s3b: what do the dotted lines in the right-hand graphs represent?

I believe the contacts for Adtech grids is 2.3 mm diameter exposed, not 4 (4 is the disc diameter, partly covered by silicone)

discussion section

Results on a single electrode can by no means be 'challenging the strict topography of the motor homunculus', it is at most a faint suggestion until this is replicated in more electrodes and subjects

The suggestion that only 1 electrode is needed for a BCI implant (p10) for imagined speech decoding is puzzling: how would one find that tiny area without opening the skull and search for it?

Authors state that they 'observed that the frequency band contributions of imagined speech are more similar to those of performed speech in the motor cortex than to perceived speech in the STG'. Yet, this is based on wilcoxon tests for individual electrodes, and no second-level analysis is shown to substantiate this observation, nor a quantitative measure of (dis)similarity

Authors also state they ‘also observed a larger overlap of electrodes between imagined and performed speech in comparison with the overlap between imagined and perceived speech.’ I appreciate that the numbers in fig 2 differ but simply counting the numbers of electrodes that exceed a threshold of detection in one or more conditions does not constitute a significance. For that, one needs to statistically compare un-thresholded detection metrics across electrodes (perhaps best limited to those electrodes that respond to any condition). One could do this for regions separately including only subjects with electrodes in such area. It is a common misconception that one can derive firm conclusions from binarizing statistical values (eg $p < 0.05$ leads to value 1, $p > 0.05$ leads to value 0) and then adding them up for multiple tests (here electrodes) to then state numbers are different for different groups or tasks. This becomes apparent if for example all electrodes in one region show an effect slightly above $P = 0.05$, and all in another region show an effect slightly below $p = 0.05$. The actual difference between the regions is marginal, but the binarizing method inflates the difference dramatically

Reviewer #2 (Remarks to the Author):

This paper trains algorithms to detect 3 types of speech based on the ECoG recordings from 16 participants measured in epilepsy patients who were acutely implanted with subdural or epidural grids and/or depth electrodes as part of their clinical workup. The authors investigate the contribution of each frequency band, the importance of each electrode, and the multi-electrode performance. Finally, the paper describes the transfer learning of the model across speech modalities and subjects.

The authors claim that one of the principal novelties of their work is the capability to detect imagined speech from ECoG recordings, which is difficult due to the lack of behavioural measurable output. On the other hand, based on single electrode analysis, the other important result is the identification of brain regions that are relevant for each speech modality and for which frequency band.

In general, the paper is well written and organised and the results are clearly presented, nevertheless I have some major comments that could help to improve the ms.

1) The figures from 2 to 7 contain a lot of information including numbers and text that are very difficult to read. Please improve the quality of the figures and enlarge the numbers and letters. In particular:

- Fig 2: The authors attempt to condense a lot of information in each circle representing each electrode, but as it is, it is impossible to differentiate the diameter of the circle and the three colors within each one representing each modality. Please, it would be important to clarify this representation, maybe to create one representation for each modality as fig 4?
- Fig 3: enlarge the letters and numbers and the quality of the panels which is notably different from the letter of the titles
- Fig4: the same
- Fig 5: the colorbar is not possible to read, include in the figure which means the colorbar, and the

frequency bands (as fig1). Maybe showing the average across the three-time instances could help to clarify the figure.

- Fig 6: quality and the size of the numbers. Also, it looks like the aspect ratio of each transfer learning accuracy square is changed.

- Fig 7: the numbers of the transfer learning information

2) It would be great if the authors could be more specific about the goal and relevance of the study. From my point of view, it would be interesting to focus on the imagined speech importance and then, as a complement of the ms, the transfer and similarities and differences with the other modalities. In that direction, it could help to include in the abstract why is important to detect and decode imagined speech

3) It would be interesting to clarify if the sentence that each participant reads and memorizes before the task is relevant or not. Is it always the same?

4) In the multi-electrode analysis, do you investigate the collinearity between the electrodes that are included in the linear regressor? The number of electrodes included in each case are informative (and should be reported, I am not sure if there are included in fig 3, because it is not possible to read it)? It would be interesting if the author could provide some information about this.

5) Considering that your findings show overlap between important electrodes between speech modalities, do you consider creating a detection model using different modalities for a single subject? It could be useful to create a model that detects speech in different modalities?

Minor:

In the method section, there are a couple of XX, evidently as placeholders. Please replace it.

Reviewers' comments:

Reviewer #1 (Remarks to the Author):

commsbiol 23-3065

1.1 The manuscript reports on detection of speech spoken, heard or imagined, using ECoG data from 16 subjects, with a focus on imagined speech. Several types of analyses are presented, to assess relationships between the three. The logic of the chosen analyses and the nature of the results is not quite convincing, it seems more of a data exploration effort than testing clear hypotheses. Given the difficulty of obtaining the valuable human ECoG data, electrode location and patient are intermixed, resulting in challenges to separate regional phenomena from individual differences. Several aspects of the presented research dampen my enthusiasm, mainly because of a lack of second-level analyses to substantiate interpretations/conclusions, and the fact half of the analyses was conducted on an extremely small selection of electrodes. Accordingly, results are not very amenable to generalizable conclusions based on quantitative assessment, for which reason I hesitate to see how this moves the field forward. Yet I do think it is possible to improve the robustness of findings when second-level analyses are added and results thereof considered. In current form, there is no clear discourse on effect size of different data analysis comparisons, nor any second-level analysis probing consistency of effects across subjects. Furthermore, half of the analyses sections involve only 7 (or even 6) electrodes out of the total pool of 588, and conclusions are drawn wide based on this very small selection. It is also not explained why these electrodes (1 per subject for 7 subjects) were chosen. Any interpretation of differences between electrodes is thus quite possibly confounded with differences between individuals (all of whom have epileptic activity on top of it). Results are discussed and interpreted as more generally applicable than analyses support, given that there are no second-level assessments across subjects. Grids understandably differ in brain region coverage across subjects but at least one could form subgroups with similar coverage on particular brain regions for this purpose, allowing for more powerful second-level (across-individual) analyses.

We understand the reviewer's concerns and thank him/her for the valuable comments. Imagined speech is often perceived as elusive to able participants, making it challenging to posit, from the start, a clear hypothesis to test. Even the nature of imagined speech (whether it relates more to performed or more to perceived speech or still to something more abstract¹) is not well understood. The goal of our paper was to investigate imagined speech to gain a better understanding of which brain regions and electrophysiological frequencies play a key role. The strategy we adopted to do so started from developing a model that detects speech events. Then, we analyzed the performance of the model across the brain and identified which

frequency bands contribute to the detection per speech mode. We reported the best event detection electrodes in the main text, but we did also verify whether including more electrodes would increase speech event detection performance (see Fig. 3 in the paper). As to the selection of 7 patients in Fig. 5: this was done to facilitate the cross-modality comparison, where we showed only those patients for whom there is no doubt that imagined speech was also detected. But we did also report the result for all patients and all 3 speech modes in the Supplementary material section as well as their functional mappings which should confirm the role of the reported electrode locations (Supplementary Fig. P1-16).

However, we agree that the manuscript could benefit from a more elaborate second-level analysis that would consider all electrodes regardless of their performances. To overcome this issue, we divided the electrodes into three groups (temporal, frontoparietal and occipital lobe) and conducted additional second-level analyses. These analyses are explained in our replies to the reviewer's comments. We also report the changes we made to the manuscript (added paragraphs and/or figures).

Comments

Methods/Results Section

1.2 Authors attribute much importance to the degree of low frequency contribution to speech detection and differences across brain regions. Yet, one possible explanation deserves consideration: results of separate bands for decoding speech regarding the gamma bands in auditory cortex may be explained by auditory attention processes increasing gamma power throughout the task, thereby increasing baseline activity and reducing any difference with further activation. This would be somewhat less in sensorimotor cortex since speed (hence preparation/readiness) is not important for participants. One could argue that the seen differences in high/low frequency contribution reflects functional differences and can therefore not be directly compared in terms of anatomical features between regions.

First of all, we note that our participants were instructed by a reading cue for all trials, so auditory attention processes could therefore only be present during listening trials. In that case, depending on the sampled region, time windows before the speech onset or after the speech offset could indeed exhibit elevated gamma activity due to auditory attention processes. If we had compared speech events to resting state rather than these windows surrounding the speech event, it would be possible that gamma band activity plays a bigger role in certain regions. However, the goal of this study was to detect *speech events*, as it would be the case in an application, by delimiting the precise boundaries of the speech events, given the influence of auditory attention or any preparatory processes that would also occur in real-life scenarios. For that purpose, low frequencies seem to play an important role.

1.3 In the comparison between speech modes section on p7, only 7 subjects and only 1 electrode per subject is analysed, out of 588 total. How is this motivated? Results are then hardly representative for a larger group, so the information obtained from this analysis seems marginal and conclusions rather overly generalized.

The 7 patients shown in Fig. 5 were those for whom there is no doubt that imagined speech could be detected. But we did also report the results for all patients and all 3 speech modes in the Supplementary material section as well as their functional mappings which should confirm the role of the reported electrode locations (Supplementary Fig. P1-16). Since imagined speech events could be detected from a few electrode locations only (as it is the case in other studies¹⁻³), we decided to shed light on them. The caption of Fig. 5 was changed to make the electrode selection procedure clearer. Note that we also examined whether including more electrodes improved event detection accuracy (Fig. 3). Figure 5 illustrates what could already be observed in Fig. 4: the gamma band contribution differs between the sensorimotor cortex and the temporal lobe for imagined speech. We now report on an extra second-level analyses to substantiate this claim, as explained further in this letter.

The caption of Fig. 5 is now:

Figure 5 - Frequency band contributions per speech mode at the best imagined speech detection electrode. The single-electrode accuracy and activation coefficients (color scale shown bottom right) are shown for 7 participants, for each speech mode (labeled by the icons), for the 7 frequency bands and 3 time instances. From our population of 16 participants, we display the results only for those participants for which the accuracy at the best imagined speech detection electrode was above chance level with $p < 0.001$ for all speech modes (see Table S2a in the supplementary material). For subject P11, the second best imagined speech electrode (corresponding to the best electrode for performed speech) is shown since the functional mapping revealed that this electrode was located in the supplementary motor area while the first best electrode was labelled as 'seizure onset' (see Fig. P11 in the supplementary material). The accuracies of the best electrodes for all subjects and speech mode can be found in the supplementary material. Dotted lines point to right hemisphere electrodes, dashed lines to left hemisphere electrodes.

1.4 Was each included electrode investigated for epileptic activity? given that grids are placed on clinically suspected epileptic zones, many electrodes could exhibit interictal phenomena contaminating the signals (hence analyses). For instance, for P3 a negative gamma response is observed for imagined speech, and virtually no response in the other 2 conditions. A negative gamma response may well be related to epileptogenic phenomena, where mental focus/attention tends to suppress the normally elevated interictal gamma power.

Our channels were checked for interictal activity by an experienced epileptologist blinded to the study results. Signals were reviewed and each channel was given a score from 1 to 3 based on the emergence of interictal activity (1 = infrequent, 2 =

moderate, 3 = frequent). The results of this analysis are now added to the supplementary material section. In sum, less than 10% of the channels exhibited frequent interictal activity (42 channels out of 588). None of the electrodes shown in Fig. 5 exhibited frequent interictal activity. In particular, all electrodes of participant P3 were labelled as level 1.

We do understand the reviewer's concern that interictal activity could be modulated by attentional processes and therefore could "help" the model. Meisenhelter et al.⁴ showed that the occurrence of interictal epileptiform discharges decreased by approximately 20% when patients were looking at a word they needed to memorize compared to when they were looking at a blank screen. However, we could not find evidence in the literature of interictal activity being consistently modulated by a cue or task (which would direct model training). Furthermore, our paradigm consists of multiple cues that requires the patient's attention throughout the trial. Finally, our model detects changes relative to the time windows before speech onset and after speech offset but not relative to resting state (see our reply to the reviewer's first comment).

In order to investigate further the relation between model performance and interictal activity, the model implemented by Quon et al. 2022⁵ was used to detect interictal epileptiform discharges (IED). The proposed model achieves a high performance to detect high-amplitude spikes (above 95%, as reported in their paper) but lower performance to detect spikes of atypical morphologies (66.15%). Furthermore, since it only detects interictal activity associated with spikes, it is therefore not reliable for all participants and visual inspection remains the gold standard. Participant P9 was chosen as his/her signal exhibited clear high-amplitude IED and multiple electrodes were labelled as level 2 and 3. The participant also reaches a high performance for imagined speech detection and one of his/her electrodes is shown in Fig 5. The correlation between the number of spikes detected by the model and the level of interictal activity attributed by the epileptologist reached a value of 0.7325, indicating a consensus between both measures. The correlation between the number of spikes detected and the model performance for imagined speech detection was strongly negative (see Fig. R1) providing evidence that channels with lots of interictal activity are unlikely to reach a high event detection accuracy.

Figure R1 – Single-electrode performance for imagined speech detection in function of the number of spikes detected by an IED detector for participant P9. Each dot corresponds to an electrode and the line depicts the regression line.

The following paragraph was added to the methods section:

No electrodes were removed due to interictal activity, given the absence of an objective criterion to do so. Nevertheless, the occurrence of interictal activity was assessed by an experienced epileptologist who provided level scores (level 1 = infrequent, level 2 = moderate, level 3 = frequent) to each channel which are reported in the supplementary material (see Fig. P1-P16). Less than 10% of the channels exhibited frequent interictal activity (42 channels out of 588). Overall, electrodes with lots of interictal activity did not reach a high speech event detection accuracy.

1.5 How was it determined that there was no residual articulator movement during imagined speech? such may well have generated sensorimotor cortex activation. We understand the reviewer’s concern. Ideally, electromyographic (EMG) signals should be recorded to detect the presence of articulator activity. However, this was not done in our study since EMG signals do not serve a clinical purpose and were thus not recorded. We therefore opted for a methodology similar to Proix et al (2022)¹.

The following paragraph was added to the paper in the subsection “Experimental design” of the Methods section:

An experimenter was present throughout the whole experiment to check that the patient understood the instruction and would not produce articulator movements. If a doubt would have persisted, we could also ask to review the video sequence since patients are video monitored for clinical purposes. Finally, we also checked the audio recordings during imagined speech trials to check that no sound was produced. Three patients were excluded from the study since they could not imagine speaking without producing a sound or moving their lips (as mentioned in the previous paragraph).

1.6 What was the nature and purpose of the passive dataset analysis? As a sort of correction it seems, it was not applied to the other task types. Thresholds for effects in this dataset seem to be arbitrary as I miss a clear rationale.

The passive dataset was used to add an extra criterion to avoid including electrodes that were not specific to imagined speech but to visual cues. This exclusion criterion was only applied to imagined speech because of the lack of behavioral output which doesn't allow us to have a precise timing of the speech events.

The following paragraph was added in the subsection "Imagined speech timing" of the Methods section:

The passive dataset was used to define an exclusion criterion to reject electrodes that are likely to respond to visual cues rather than to imagined speech events. If the accuracy on the passive dataset was below 50%, the overall performance was set to 50% or the electrode was excluded from the analysis depending on the context.

1.7 Was there an actual benefit from including separate time epochs (3 of 125 ms consecutive), given the graphs (fig 5 and 7) indicate virtually no differences?

Collapsing these 3 timepoints could add some power to analyses.

Yes, there was a benefit to take separate time epochs as features. As can be seen in the figure below, we analyzed how the number of points in the time window affected model performance and observed that taking 3 timepoints was optimal.

Figure R2 – Single-electrode performance for performed speech in function of the number of time samples taken in the 250 ms window (65 corresponding to all time points, given the 256 Hz sampling rate) for one electrode from patient P1.

1.8 fig 4: color coding is rather unclear (same basecolor for opposite effects), consider using different colors for larger/equal/smaller

1.9 fig s3b: what do the dotted lines in the right-hand graphs represent?

1.10 I believe the contacts for Adtech grids is 2.3 mm diameter exposed, not 4 (4 is the disc diameter, partly covered by silicone)

Thank you. We have accommodated these suggestions.

discussion section

1.11 Results on a single electrode can by no means be 'challenging the strict topography of the motor homunculus', it is at most a faint suggestion until this is replicated in more electrodes and subjects

We agree that the wording is too strong and we removed it from the text. This suggestion was based on electrodes from two subjects and previous studies that showed speech activity within the hand motor region^{6,7}. Our results only add evidence but do not prove or validate our assertion.

1.12 The suggestion that only 1 electrode is needed for a BCI implant (p10) for imagined speech decoding is puzzling: how would one find that tiny area without opening the skull and search for it?

We understand the confusion since it would indeed not be an easy task to find the exact spot where speech detection is optimal. What we meant was that low-density electrocorticography (ECoG) might be sufficient for speech detection (while high-density ECoG seems more important for speech decoding) and that speech detection itself would not require many electrodes to be chronically implanted. Moreover, strips of electrodes might be sufficient and these can be placed through burr holes. This was done in Vansteensel et al. (2016)⁸ where they placed 4 strips of 4 electrodes at locations determined with the use of fMRI. They could detect attempted movements from a pair of electrodes (i.e. bipolar referencing, we used common average referencing).

The following end of paragraph was changed in the discussion section:

Because of this, the multi-electrode model improved the imagined speech performance for only two participants while more participants benefitted from multiple electrodes for performed and perceived speech. This suggests that, only few electrodes might be sufficient to chronically implant for imagined speech detection and that high-density ECoG might not be required. This is in line with Vansteensel et al.⁸ who implanted a patient with late-stage amyotrophic lateral sclerosis with low-density strips of subdural electrodes through burr holes. They could detect movement attempts using only one pair of electrodes (i.e., one signal given bipolar referencing). The location of the electrodes had been determined with the use of fMRI and anatomical landmarks. A similar procedure could be applied to detect speech attempts.

1.13 Authors state that they 'observed that the frequency band contributions of imagined speech are more similar to those of performed speech in the motor cortex than to perceived speech in the STG'. Yet, this is based on wilcoxon tests for individual electrodes, and no second-level analysis is shown to substantiate this observation, nor a quantitative measure of (dis)similarity.

Second-level analysis is limited due to the small number of electrodes where imagined speech events could be detected. Hence, as suggested by the reviewer, we

subdivided our electrodes into three groups (temporal, frontoparietal and occipital lobe) and conducted several analyses based on this subdivision. The figures that summarize the results of these analyses were added in the supplementary material section.

The next paragraph was added to the subsection "Speech detection performance" of the results section:

Additionally, we noted that, although less electrodes were placed in the left temporal lobe, more electrodes reached a high accuracy for imagined speech detection in the left compared to the right temporal lobe (see Supplementary Table S2b). A Wilcoxon rank-sum test confirmed that the accuracy in the left temporal lobe is larger than in the right temporal lobe (p -value= $2.34e-05$, alternative: accuracy in the left temporal lobe is larger than in the right temporal lobe, $N=140$ in the right temporal lobe and $N=98$ in the left temporal lobe).

The next paragraph was added to the subsection "Full spectrum vs. low-frequency and gamma band speech detection" of the results section:

We conducted a Wilcoxon signed-rank test to compare the performance using only low frequency versus only high frequency in the temporal lobe (alternative: accuracy using only low frequency > only high frequency) and the paired test was found to be significant ($p < 1e-7$, $N=238$) for all speech modes, confirming the importance of lower frequencies in the temporal lobe.

The following paragraph was added to the subsection "Comparison between speech modes" of the results section and figures were added in the supplementary material: *In order to further investigate the relation between speech modes and brain regions, we subdivided electrodes into three groups (temporal, frontoparietal and occipital regions) and conducted a few extra analyses based on this subdivision. Electrodes from the 10 subjects who had at least one electrode performing above chance level for imagined speech detection were included in this analysis (143 electrodes in temporal lobes and 232 electrodes in the frontoparietal region, see Table S2a). We defined as "gamma activation" the average of the activation pattern coefficients of low-gamma and high-gamma bands. We then compared the gamma activation for performed and imagined speech in both the temporal and frontoparietal regions. We observed a significant difference between the gamma activation of performed and imagined speech in the temporal lobe (Wilcoxon signed-rank test, $p=1.42e-09$, alternative: gamma activation of imagined speech < performed speech, $N=143$) while the p -value was much larger in the frontoparietal region ($p=0.0377$, $N=232$, see Fig. S5a). To further investigate the difference between performed and imagined speech, we also conducted a statistical test to compare the drop in gamma activation from performed to imagined speech in both regions. The decrease in gamma activation was significantly larger in the temporal lobe than in the frontoparietal region (Wilcoxon rank-sum test, $p=1.29e-04$, alternative: difference in the temporal lobe > difference in the frontoparietal region, see Fig. S5b).*

Finally, we also analyzed the relation between gamma activation and model accuracy in the temporal lobe (see Fig S5c). The correlation was positive for both performed and perceived speech while it was negative for imagined speech, further indicating that gamma bands play a less important role in the temporal lobe for imagined speech detection.

Supplementary Figure S5a – **Gamma activation comparison.** Distribution of gamma activation across the electrodes for performed and imagined speech in the temporal lobe and the frontoparietal region. Electrodes from the 10 participants with at least one electrode performing better than chance level for imagined speech detection were included (143 electrodes in temporal lobes and 232 electrodes in the frontoparietal region). The difference between the gamma activation of performed and imagined speech was tested in both the temporal lobe and the frontoparietal region (Wilcoxon signed-rank test, alternative: gamma activation of imagined speech < performed speech, $p=1.42e-09$ and $p=0.0377$, respectively).

Supplementary Figure S5b – **Gamma activation difference between speaking and imagining.** Distribution of the difference between imagined and performed speech across electrodes in the temporal lobe and frontoparietal region. Electrodes from the 10 participants with at least one electrode performing better than chance level for imagined speech detection were included (143 electrodes in temporal lobes and 232 electrodes in the frontoparietal region). The difference was found to be larger in the temporal lobe than in the frontoparietal region (Wilcoxon rank-sum test, $p=1.29e-04$, alternative: difference in the temporal lobe > difference in the frontoparietal region).

Supplementary Figure S5c – **Relation between gamma activation and model accuracy in the temporal lobe.** Each dot corresponds to an electrode in the temporal lobe. Electrodes from the 10 participants with at least one electrode performing better than chance level for imagined speech detection were included (143 electrodes in temporal lobes). The line depicts the regression line. Correlation values were 0.5078, 0.3894 and -0.2340 for performed, perceived and imagined speech respectively.

1.14 Authors also state they 'also observed a larger overlap of electrodes between imagined and performed speech in comparison with the overlap between imagined and perceived speech.' I appreciate that the numbers in fig 2 differ but simply counting the numbers of electrodes that exceed a threshold of detection in one or more conditions does not constitute a significance. For that, one needs to statistically compare un-thresholded detection metrics across electrodes (perhaps best limited to those electrodes that respond to any condition). One could do this for regions separately including only subjects with electrodes in such area. It is a common misconception that one can derive firm conclusions from binarizing statistical values (eg $p < 0.05$ leads to value 1, $p > 0.05$ leads to value 0) and then adding them up for multiple tests (here electrodes) to then state numbers are different for different groups or tasks. This becomes apparent if for example all electrodes in one region show an effect slightly above $P = 0.05$, and all in another region show an effect slightly below $p = 0.05$. The actual difference between the regions is marginal, but the binarizing method inflates the difference dramatically.

We understand the issue raised by the reviewer. We followed a similar procedure as was done in Soroush et al. 2023² to which we refer and compare our results. We now provide an extra analysis in the supplementary material that compares un-thresholded metrics (see Fig. S2).

This paragraph was added at the end of subsection "Speech detection performance" of the Results section:

We also analyzed the correlation between the performance of imagined speech detection with performed and perceived speech for all participants (see Fig. S2). In both cases, we observe that imagined speech performance is positively correlated with performed and perceived speech performance (correlation of 0.4253 and 0.4086, respectively). However, when looking at the best electrodes only (performance above 65%), we observed that the trend is reversed for perceived speech (correlation becomes -0.3730), meaning that the best electrodes for perceived speech detection do not necessarily match with the best electrodes for imagined speech detection.

Supplementary Figure S2 – Relation between speech mode accuracies. Imagined speech accuracy is plotted as a function of the performed (panel a) and perceived (panel b) speech accuracy. Each dot corresponds to an electrode (all electrodes from all participants were included in this analysis). The solid lines depict the regression lines across all points (correlation of 0.4253 and 0.4086 for panel a and b, respectively), while the dashed lines depict the regression lines for electrodes whose accuracies for both speech modes are above 65% (correlation of 0.1975 and -0.3730 for panel a and b, respectively).

Reviewer #2 (Remarks to the Author):

This paper trains algorithms to detect 3 types of speech based on the ECoG recordings from 16 participants measured in epilepsy patients who were acutely implanted with subdural or epidural grids and/or depth electrodes as part of their clinical workup. The authors investigate the contribution of each frequency band, the importance of each electrode, and the multi-electrode performance. Finally, the paper describes the transfer learning of the model across speech modalities and subjects. The authors claim that one of the principal novelties of their work is the capability to detect imagined speech from ECoG recordings, which is difficult due to the lack of behavioural measurable output. On the other hand, based on single electrode analysis, the other important result is the identification of brain regions that are relevant for each speech modality and for which frequency band.

In general, the paper is well written and organised and the results are clearly presented, nevertheless I have some major comments that could help to improve the ms.

We thank the reviewer for the positive comment and took into consideration the suggested improvements.

2.1 The figures from 2 to 7 contain a lot of information including numbers and text that are very difficult to read. Please improve the quality of the figures and enlarge the numbers and letters. In particular:

- Fig 2: The authors attempt to condense a lot of information in each circle representing each electrode, but as it is, it is impossible to differentiate the diameter of the circle and the three colors within each one representing each modality. Please, it would be important to clarify this representation, maybe to create one representation for each modality as fig 4?
- Fig 3: enlarge the letters and numbers and the quality of the panels which is notably different from the letter of the titles
- Fig4: the same
- Fig 5: the colorbar is not possible to read, include in the figure which means the colorbar, and the frequency bands (as fig1). Maybe showing the average across the three-time instances could help to clarify the figure.
- Fig 6: quality and the size of the numbers. Also, it looks like the aspect ratio of each transfer learning accuracy square is changed.
- Fig 7: the numbers of the transfer learning information

We apologize for the resolution of the figures. The conversion to pdf file on the uploading platform decreased drastically the resolution of the figures. The resolution as well as the size of numbers and letters were increased in the revised manuscript. The size of Fig. 2 was increased to improve readability. Note that the information in Fig. 2 can also be found in Fig. 4 ("full spectrum" column). We added the following sentence in the legend of Fig. 2:

The results for panel (a) and (b) can also be viewed in the first column ("Full spectrum") of Fig. 4 for each speech mode separately.

2.2 It would be great if the authors could be more specific about the goal and relevance of the study. From my point of view, it would be interesting to focus on the imagined speech importance and then, as a complement of the ms, the transfer and similarities and differences with the other modalities. In that direction, it could help to include in the abstract why is important to detect and decode imagined speech
The introduction and abstract were changed to better highlight the importance of imagined speech.

This sentence was added to the abstract:

Nevertheless, imagined speech is advantageous since it does not depend on any articulator movements that might become impaired or even lost throughout the stages of a neurodegenerative disease.

This text was added in the first paragraph of the introduction:

Even though recent progress has been made with patients unable to produce intelligible speech, it remains a challenge to help those who completely lack residual movements, such as locked-in syndrome patients⁹. For patients suffering from neurodegenerative diseases, imagined speech would not depend on the stage of the disease which is a serious advantage compared to models relying on "attempted speech" and thus residual articulatory movements which might gradually disappear.

2.3 It would be interesting to clarify if the sentence that each participant reads and memorizes before the task is relevant or not. Is it always the same?

A set of 20 sentences was selected which can be found in the supplementary material. Although they don't relate to a clinical context, they were controlled for the number of syllables and complexity and span a wide vocabulary range¹⁰.

2.4 In the multi-electrode analysis, do you investigate the collinearity between the electrodes that are included in the linear regressor? The number of electrodes included in each case are informative (and should be reported, I am not sure if there are included in fig 3, because it is not possible to read it)? It would be interesting if the author could provide some information about this.

Yes, the collinearity between the electrodes was investigated. We also used a ridge regression model instead of regular linear regression to account for collinearity, but the performance did not improve. This might be due to low-density EcoG grids exhibiting less collinearity compared to high-density ones. In the end, the multi-electrode model comprises two steps: first, models are built for each electrode separately and, second, the output of all these models were combined into a single value. The numbers of electrodes are included in Fig 3, but we have now increased the font size to make them better readable.

2.5 Considering that your findings show overlap between important electrodes between speech modalities, do you consider creating a detection model using different modalities for a single subject? It could be useful to create a model that detects speech in different modalities?

This is indeed an interesting line of research, which we now added to the discussion section. In this paper, we mainly focused on the brain regions and frequencies relevant for each speech mode and thus this motivated having one model per speech mode. But surely, one model that detects all speech modes would be relevant for an application.

The following sentence was added to the discussion:

Developing a single model that operates across the three speech modes is an interesting line of future research.

Minor:

2.6 In the method section, there are a couple of XX, evidently as placeholders. Please replace it.

This was done on purpose to preserve the anonymity of the authors during the peer review process.

References

1. Proix, T. *et al.* Imagined speech can be decoded from low- and cross-frequency intracranial EEG features. *Nat Commun* **13**, 48 (2022).
2. Soroush, P. Z. *et al.* The nested hierarchy of overt, mouthed, and imagined speech activity evident in intracranial recordings. *Neuroimage* **269**, 119913 (2023).
3. Pei, X., Barbour, D. L., Leuthardt, E. C. & Schalk, G. Decoding vowels and consonants in spoken and imagined words using electrocorticographic signals in humans. *J Neural Eng* **8**, 046028 (2011).
4. Meisenhelter, S. *et al.* Interictal Epileptiform Discharges are Task Dependent and are Associated with Lasting Electrocorticographic Changes. *Cereb Cortex Commun* **2**, (2021).
5. Quon, R. J. *et al.* AiED: Artificial intelligence for the detection of intracranial interictal epileptiform discharges. *Clinical Neurophysiology* **133**, 1–8 (2022).
6. Stavisky, S. D. *et al.* Neural ensemble dynamics in dorsal motor cortex during speech in people with paralysis. *Elife* **8**, e46015 (2019).
7. Wilson, G. H. *et al.* Decoding spoken English from intracortical electrode arrays in dorsal precentral gyrus. *J Neural Eng* **17**, 066007 (2020).
8. Vansteensel, M. J. *et al.* Fully Implanted Brain–Computer Interface in a Locked-In Patient with ALS. *New England Journal of Medicine* **375**, 2060–2066 (2016).
9. Ramsey, N. F. & Crone, N. E. Brain implants that enable speech pass performance milestones. *Nature* **620**, 954–955 (2023).
10. van Wieringen, A. & Wouters, J. LIST and LINT: Sentences and numbers for quantifying speech understanding in severely impaired listeners for Flanders and the Netherlands. *Int J Audiol* **47**, 348–355 (2008).

REVIEWERS' COMMENTS:

Reviewer #1 (Remarks to the Author):

I appreciate the responses and additional analyses very much and find them satisfactory for which I thank the reviewers.

I do have one remaining issue with imagined speech: given that the authors place the work in the context of decoding in completely paralyzed people, the lack of adequate measures to exclude any non-perceivable movements (eg subliminal vocal cord contractions) in these abled participants poses a potential confound. I appreciate that it was not feasible to record any EMG in these patients. Yet, I believe a comment on this potential confound should be added in the discussion as a study limitation, ie that one cannot exclude some movement and therefore somatosensory input to the cortex that may contribute to decoding imagined speech events.

Reviewer #2 (Remarks to the Author):

The authors have addressed all my comments. I thank authors for the efforts. I find the current version of the ms suitable for publication

Reviewer #1 (Remarks to the Author):

I appreciate the responses and additional analyses very much and find them satisfactory for which I thank the reviewers.

I do have one remaining issue with imagined speech: given that the authors place the work in the context of decoding in completely paralyzed people, the lack of adequate measures to exclude any non-perceivable movements (eg subliminal vocal cord contractions) in these abled participants poses a potential confound. I appreciate that it was not feasible to record any EMG in these patients. Yet, I believe a comment on this potential confound should be added in the discussion as a study limitation, ie that one cannot exclude some movement and therefore somatosensory input to the cortex that may contribute to decoding imagined speech events.

Reviewer #2 (Remarks to the Author):

The authors have addressed all my comments. I thank authors for the efforts. I find the current version of the ms suitable for publication.

We thank the reviewers for their valuable comments which helped improve the manuscript. We modified the last paragraph of the discussion section accordingly.

While the reach of this study was limited due to the restricted time for our experiments and the low-density electrode coverage, we believe our findings contribute to the implementation of imagined speech decoding pipelines. We acknowledge that, despite subjects were asked to refrain from any movement, non-perceivable muscle activity, even subliminal (e.g., vocal tract activity), cannot be ruled out without simultaneous electromyographic monitoring. Nevertheless, our results suggest that ECoG frequency bands in the temporal lobe and motor cortex contribute differently to speech events but consistently across subjects and speech modes, which could be exploited to pre-train components of the speech decoding pipeline.